# LAP-BFT: Lightweight Asynchronous Provable Byzantine Fault-Tolerant Consensus Mechanism for UAV Network

**Lingjun Kong** , **Bing Chen** * and **Feng Hu**

The College of Computer Science and Technology, Nanjing University of Aeronautics and Astronautics, Nanjing 210016, China; lingjun@nuaa.edu.cn (L.K.); huf@nuaa.edu.cn (F.H.)
* Correspondence: cb_china@nuaa.edu.cn

**Abstract:** Mission-oriented UAV networks operate in nonsecure, complex environments with time-varying network partitioning and node trustworthiness. UAV networks are thus essentially asynchronous distributed systems with the Byzantine General problem, whose availability depends on the tolerance of progressively more erroneous nodes in the course of a mission. To address the resource-limited nature of UAV networks, this paper proposes a lightweight asynchronous provable Byzantine fault-tolerant consensus method. The consensus method reduces the communication overhead by splitting the set of local trusted state transactions and then dispersing the reliable broadcast control transmission (DRBC), introduces vector commitments to achieve multivalue Byzantine consensus (PMVBA) for identity and data in a provable manner and reduces the computational complexity, and the data stored on the chain is only the consensus result (global trustworthiness information of the drone nodes), avoiding the blockchain's "storage inflation" problem. This makes the consensus process lighter in terms of bandwidth, computation and storage, ensuring the longevity and overall performance of the UAV network during the mission. Through QualNet simulation platform, existing practical asynchronous consensus algorithms are compared, and the proposed method performs better in terms of throughput, consensus latency and energy consumption rate.

**Keywords:** mission-oriented UAV network; Byzantine fault-tolerant; lightweight asynchronous provable consensus

## 1. Introduction

The use of UVAs as a flight platform is growing rapidly. Due to their inherent attributes such as mobility, flexibility and adaptive altitude, UAVs have many key potential applications in wireless systems [1]. UAV networks performing missions in clusters of UAVs have irreplaceable applications in emergency networking, rescue and military applications due to their light weight and fast deployment, such as [2] the study of the phase synchronization problem of establishing connections between base stations (BS) and ground receivers (GR) by a group of UAVs as relays. The UAV network is a mobile ad hoc network that is not supported by a reliable central authority and the completion of the mission relies on the interoperability of the UAV nodes. Maintaining the trustworthiness of the UAV network and correctly assessing the reliability and trustworthiness of the nodes is therefore key to mission accomplishment. However, the complex mission environment exposes the UAV network not only to network partitioning caused by physical interference but also to the risk of malicious cyber attacks from external nodes. Moreover, during the mission, legitimate drone nodes can become faulty or selfish due to external interference and energy consumption; the open nature of wireless networks also makes drone nodes more vulnerable to network attacks (e.g., link layer attacks) and compromises them to become Byzantine nodes with a legitimate identity. The absence of central authoritative support, the dynamic generation of errant nodes and the arbitrary nature of Byzantine node behavior make the UAV network during a mission essentially an asynchronous distributed system in a Byzantine environment. Information passed between nodes may be

discarded, delayed or even tampered with. It is therefore necessary to establish a highly trusted distributed Byzantine fault-tolerant system for the UAV network, thus ensuring high availability for resource-constrained UAV networks in unfriendly mission environments. The key to solving the problem lies in sensing node state changes in real time, accurately identifying untrustworthy nodes and isolating them from the mission network in time for the UAV network to effectively reach consensus on the latest state records of all participating nodes. In this regard, the Byzantine environment, the asynchronous nature of the UAV network and resource constraints are the main challenges in establishing a lightweight and efficient consensus-based UAV trusted network.

Distributed systems rely on message passing to enable communication and coordination between processes or nodes, and consensus algorithms are key to achieving data consistency among system components. Therefore, consensus algorithms have been a hot research topic in distributed systems and are the core of blockchain. According to the fault tolerance of distributed systems for faulty components, consensus protocols are divided into two main categories, namely, crash-tolerant protocols (CFT) and Byzantine fault-tolerant protocols (BFT) [3]. The consensus protocol for the UAV network explored in this paper is a BFT protocol that needs to resist not only malicious attacks from external nodes but also faces interference from dynamically generated faulty nodes internally. The consensus algorithm is divided into two main steps: first, the selection of the master node and the determination of the proposed master node; second, the agreement of consensus on the proposal. The consensus protocol is divided into deterministic and probabilistic consensus based on the consistency decision. Castro and Liskov [4] first proposed the practical Byzantine consensus algorithm (PBFT) based on replication technology, which first made the implementation of highly available distributed fault-tolerant systems possible. The PBFT specifies that all nodes take turns to be the master node and uses a three-stage protocol (sorting, communication and acknowledgment) by two-by-two communication between nodes to achieve deterministic consensus, with good consensus efficiency and no possibility of changing the consensus result. However, the identity of participants must be clear, the communication complexity is high, and the scale of the system is limited. Its application scenarios usually have relatively sufficient bandwidth and computing power, so it is widely used in coalition chains of manageable scale. However, the high complexity of communication, the need for relatively sufficient bandwidth and computing power and the need to clearly identify system participants result in limited system size and application scenarios. As an unlicensed public chain, Bitcoin uses "Satoshi Nakamoto Consensus", essentially a probabilistic consensus that competes for master nodes through proof-of-work (POW), sends new blocks through only one round of broadcasts and uses multiple confirmations to progressively increase the probability of consistency. The literature [5,6] uses POW consensus-only blockchain combined with MANET to assist in establishing trusted routes. The advantages are reduced communication complexity and no licensing required for participating nodes, thus increasing the scalability of the system. The disadvantages are also obvious: the time cost of consensus is too high, and the consensus result is uncertain.

The FLP Impossibility Conclusion [7] is the most fundamental conclusion of consensus algorithms (protocols) for asynchronous systems: there is no deterministic consensus algorithm that can solve the Byzantine consensus problem in an asynchronous environment, even when only benign errors occur. For specific asynchronous distributed system applications, the design of asynchronous Byzantine consensus algorithms needs to consider how to break the limits of the FLP's impossible conclusion. Mission-oriented UAV networks are lightweight and have nodes with limited bandwidth, computational power and energy availability, which, together with the presence of external malicious nodes and internal Byzantine nodes, makes the UAV network of distributed mission systems asynchronous most of the time. The complexity of the mission environment, the generation of errant nodes and the dynamics of the network topology all affect the security and activeness of the UAV network distributed system consensus mechanism. At the same time, the time-sensitive nature of the task requires improving the consensus efficiency of the algorithm

to meet the availability and trustworthiness of the UAV network. Existing asynchronous Byzantine consensus algorithms [8–10] do not set limits on the computational power of nodes in these application scenarios, although they fully consider asynchronous features (e.g., introducing random methods to save activities and eliminating cycle synchronization and network timeout settings). Given the light weight of UAV network nodes, the high mission timeliness and the asynchronous nature of the network, the key to ensuring the overall trustworthiness of the network in an environment where erroneous nodes are gradually increasing is the ability of the UAV network to quickly and efficiently agree on the latest trustworthy state of the UAV nodes involved in the mission.

The main contributions of this work are as follows:

- First, a node state blockchain monitoring system is introduced to achieve real-time scoring of neighboring nodes' forwarding behavior. Each node generates local state transactions, which contain neighboring nodes' trustworthiness loss assessment, remaining energy and neighboring nodes list. The blockchain system stages a consensus on the set of local state transactions, calculates the global trustworthiness of nodes, marks untrustworthy nodes and elects new authorized nodes. The consensus result is used as new block information to update the blockchain, and the drone nodes reconfigure the trusted network based on the latest block.
- Secondly, a fuzzy K-Modes clustering algorithm that can handle classification type data is introduced to divide the UAV network into *k* subnetwork regions based on the location of UAV node distribution, with a list of neighboring node IDs as the feature vector, and calculate the subregion center nodes. The region centers are stored in the new blocks as part of the configuration information to guarantee the maximum coverage of the UAV network in the new round of upper-layer networks and optimize the interzone routing.
- Third, the lightweight asynchronous provable Byzantine fault-tolerant algorithm (LAP-BFT) is proposed. This algorithm partitions the set of local state transactions so that each delegated authorization node is only burdened with a small share of data for reliable broadcast transmission. Multivalued Byzantine asynchronous consensus is accomplished through an external proof smart contract (with computational complexity $O(1)$) in the Genesis block. This not only reduces the bandwidth requirement for individual nodes but also avoids the high computational power consumption caused by threshold key operations. It enables asynchronous consensus to operate lightly in resource-constrained UAV networks.

The rest of paper is organized as follows: Work related to the consensus problem of the Byzantine system is in Section 2. The system model, including the network model, the threat model of the network and the structure of the blockchain, is placed in Section 3. A specific description of the recommended scheme, including blockchain node state detectors, is in Section 4. The proof and analysis of the nature of the system is presented in Section 5, where security and activity are proven and analyzed. Section 6 observes the consensus latency, throughput and energy consumption rate of each of the four asynchronous consensus algorithms by comparing their operation in a UAV network scenario. Section 7 concludes and looks at the possibility of the dynamic execution of smart contracts in UAV network trusted systems.

## 2. Related Work

Consensus as a fundamental problem in distributed computing was first introduced by Shostak, Pease and Lamport [11]. Since then, many different variants of the consensus problem have emerged to be studied in depth, e.g., [12,13]. The consensus on the latest status of nodes in a mission process by a trusted system of a UAV network belongs to the consensus problem of asynchronous Byzantine distributed systems. The consensus process consists of two parts, the determination of the nodes that propose values (called the chosen master) and the consensus protocol (the way to reach consensus). According to whether the consensus result is certain or not can be divided into definite consensus

protocol and probabilistic consensus. The emergence of bitcoin in 2008 has triggered a great deal of attention to blockchain technology, and most of the unlicensed public blockchains use probabilistic consensus method; the consensus process only requires one round of broadcast communication, and the consensus of an asynchronous Byzantine system is achieved by making the consensus probability converge to 1 through multiple rounds of confirmation of transactions, but the result is uncertain and the consensus inefficient. For the UAV network trustworthy system discussed in this paper, the consensus object is the dynamically changing status evaluation value of the running nodes, rather than the third-party customer transactions unrelated to the nodes, which has high timeliness, and the UAV network is relatively small in scale and more vulnerable to 51% attack, so the multiround probabilistic consensus cannot meet the consensus requirements of the UAV network trustworthy system. The Byzantine Fault Tolerance (BFT) protocol achieves deterministic consensus through message interactions between all nodes in the authentication environment. However, the number of erroneous nodes present in the system cannot exceed 1/3 of the total number of system nodes and the communication complexity is as high as $O(N^3)$.

The PBFT [7] is a simple and efficient Byzantine consensus scheme proposed by Castro and Liskov in 1999, and it is the first state machine that can operate correctly in an asynchronous Byzantine error-ridden scenario. The PBFT employs a number of optimizations to improve system performance, such as using message authentication codes instead of signatures for message authentication and transferring message hashes wherever possible to avoid transferring large message originals. However, the environment is vulnerable to a class of delayed attacks against the leader in a fully asynchronous network environment, rendering the system inoperable [14,15]. The consensus object of the UAV network blockchain system is the state transaction data submitted by all nodes, and as the size of the UAV network increases, so does the volume of data transmission, which is overwhelming for individual drone nodes with limited bandwidth.

Due to the well-known FLP impossibility theory, there can be no deterministic consensus in an asynchronous setting as long as one node crashes. Research on asynchronous BFT has thus long focused on theoretical limitations and feasibility. The weaker asynchronous common subset (ACS) proposed by Ben-Or [16] and Rabin [17] pioneered circumventing that impossibility through randomization. Bracha proposed a Byzantine protocol for asynchronous networks in 1987, in which he first proposed the idea of "restricting adversary behavior with a broadcast protocol before consensus" and gave the first implementation of a reliable broadcast protocol (RBC) [18]. This construction idea has had a profound impact on subsequent research on Byzantine protocols, but the scheme itself is slow to achieve consensus, and the desired number of rounds required to achieve consensus is related to the total number of nodes in the system, N, which cannot be guaranteed to be achieved within a constant number of rounds. These pioneering works have inspired many in-depth studies on asynchronous binary protocols (ABA) (which consider each node's input to be just one bit). ABA protocols have become an important part of building mature BFT or atomic broadcast protocols [19–22], but experiments in [8] show that running a large number of ABA instances becomes a bottleneck in consensus efficiency. The multivalued Byzantine Agreement system (MVBA) proposed by Cachin et al. [23] is a solution for distributed consensus. Distinguished from the 0–1 Byzantine consensus scheme (ABA) [15,24], MVBA is a multivalued consensus scheme and can provide stronger functionality. MVBA can run in a fully asynchronous network environment, and it uses the design ideas and modular design approach proposed in [10], where each node first transmits the proposed values via a broadcast protocol, and then the system runs the ABA protocol to reach consensus on the proposed values. MVBA can achieve consensus in a constant number of rounds, and the transmission cost of consensus message m is $O(N^3|m|)$. More research on asynchronous consensus is unfolding in different applications [22,23,25,26] to address the respective problems, but there are still problems of inefficient protocols, high communication complexity (up to $O(n^2)$ or even $O(n^3)$) and high computational overhead, making the performance of

these protocols drop dramatically when the system scales up, and thus difficult to enter the practical usable stage.

Miller et al. constructed an efficient Byzantine protocol, HoneyBadger BFT (HB-BFT) [8] based on the MVBA framework, using a carefully chosen underlying protocol, which uses the efficient RBC protocol from the literature [27] for the broadcast of proposed values and innovatively combines the idea of "apportionment" with an asynchronous common subset (ACS) protocol [25,28] to reduce transmission costs. The HoneyBadger BFT can achieve consensus in a constant number of rounds, and the consensus message $m$ is transmitted at a cost of only $O(N|m|)$. Due to the FLP impossibility, ABA must be a randomized protocol. This introduces the following drawback: while the expected number of "rounds" of each ABA protocol has a constant number of "rounds", the expected number of rounds to run $n$ concurrent ABA sessions can be huge, at least $O(\log_{10} n)$. More than that, these ABA instances are not executed in a fully concurrent manner. The reasons for this are firstly that not all instances start at the same time and some may start later because the input (from the previous RBC) has not yet been delivered, and secondly, that normal nodes also suffer from a drop in efficiency when faced with large concurrent executions (not enough CPU cores, etc.). When $n$ becomes large and the network is unstable, this leads to a difficult determination of the ACS runtime for HB-BFT. This is not applicable for a trusted system of a UAV network that uses the latest trusted status of UAV nodes in the runtime phase as the consensus object. The implementation of [9] confirms that the ABA protocol in HB-BFT has a greater practical impact on system performance.The time cost of running multiple ABA instances per node dominates HB-BFT through statistics on the average RBC and ABA runtimes. This pattern becomes more pronounced as the system grows in size, and the use of preferred agents, provably reliable broadcast protocols, etc., effectively reduces the number of ABA instances and speeds up the efficiency of asynchronous Byzantine consensus. The Dumbo-BFT enhances the improvement of the communication model based on [8] by proposing an optimized multivalued verified Byzantine asynchronous consensus algorithm, which greatly reduces the communication volume and reduces the communication bits from $O(\mu n^2 + \lambda n^2 + n^3)$ to $O(\mu n + \lambda n^2)$, ($\mu$ is the length of the message, $n$ is the number of nodes, indicating the network size, and $\lambda$ is the security parameter), reaching optimal performance at $\mu > \lambda n$. Refs. [8–10] present an asynchronous consensus algorithm that is practical in Byzantine asynchronous environments and is based on the optimization and improvement of the ACS protocol, but the scenarios in which it is applied do not consider the limitation of computing power, the network topology is relatively stable and the consensus data are third-party customer transactions, independent of the nodes involved in the consensus. Table 1 compares the performance of the above consensus methods, where $\lambda$ is the length of the security parameter and $N$ is the network size.

Practical asynchronous consensus algorithms have been better used. However, their threshold signature, encryption and decryption require high computing power support. The consensus object is the transaction set of a third party, which can be transmitted by randomly selecting the transaction set and decentralized by corrective code (RS_Code) encoding. However, these asynchronous consensus algorithms cannot be applied to the UAV network application scenario discussed in this paper, where the set node resources are limited and the consensus object is the state change in the node itself. Thus, the asynchronous consensus algorithm recommended in this paper replaces the threshold encryption with an external proof smart contract of computational complexity $O(1)$ with the support of an authenticated blockchain and establishes decentralized lightweight reliable broadcast transmission (LD-RBC) based on delegated authorized nodes to avoid RS_Code operations. This reduces the bandwidth pressure on nodes and improves the throughput and consensus efficiency. Additionally, the periodic update of delegated agents for consensus also effectively improves the overall fault tolerance of the UAV network.

**Table 1.** Consensus method performance comparison.

| Categories | | Papers | Overhead | | Commu. Complexity | Deterministic | Tolerance | Efficiency |
|---|---|---|---|---|---|---|---|---|
| | | | Comp | Stor | | | | |
| Sync (BFT) | | [7,11–13] | High | Low | $O(N^2) - O(N^3)$ | Yes | 1/3 | Low |
| Theoretical Asyn | | [19–26] | Mid | Low | $O(N^2)$ | Yes | 1/3 | low |
| Practical Asyn (Blcok-chain) | POW | [5,6] | Very High | High | $O(1)$ | No | 1/2 | low |
| | HB-BFT | [7] | High | High | $O(N^2|m| + \lambda N^3 \log^N)$ | Yes | 1/3 | Mid |
| | Dumbo-1 | [8] | Mid | High | $O(N^2|m| + \lambda N^3 \log^N)$ | Yes | 1/3 | Mid |
| | Bumbo-2 | [9] | Low | High | $O(N^2|m| + \lambda N^3 \log^N)$ | Yes | 1/3 | High |

## 3. System Mode

The mission-oriented UAV network consists of $N$ lightweight UAVs, denoted as $\overrightarrow{U^i}_{i \in \{1,2,...,N\}}$, and can be considered as a P2P virtual network based on a mobile ad hoc network. The global trust platform for the UAV network is a private permission blockchain system. The system is based on the elliptic curve cryptosystem of setting the public and private keys of the drone $U^i$, the drone node identity $ID_i$ and the proof of the node's existence in the network $W\_i$. The identity vector commitment and the registered node base information of the UAV network are stored in the Genesis block and synchronized to all registered UAVs before the mission starts. The security environment is set up with an elliptical public key cryptosystem, where the registration server allocates public and private keys for UAVs and generates a unique identity for UAVs by hashing the IP and public key of UAVs and mapping them to a point of the elliptical curve (finite exchange group $G$). The system constructs UAV identity vector commitment and provides a witness of existence for all registered UAVs. A random set of UAV nodes are selected as delegated agents responsible for the first round of consensus at the start of the mission. This part is not the focus of this paper, so only the setup and security foundation of the UAV network is briefly explained. The following description is the basic setup required for the blockchain system to operate. $H_1 = (0,1)^* \to Z_q^*$, $H_2 = (0,1)^* \to G$ are the hash functions to generate UAV identity,

$$ID_i = H_2(H_1(PK_i \parallel U_i) \parallel Sign_{reg}^{SK}(H_1(PK_i \parallel IP_i))) \tag{1}$$

where $Sign_{reg}^{SK}$ is the signature function of the registration server. $\overrightarrow{ID} = \{ID_1, ID_2, \ldots, ID_N\}$ is the identity vector of all registered UAVs, using the vector commitment algorithm to generate the identity vector commitment ($VC_{ID}$) and the identity witness ($W_i$) of the UAVs. $VC_{ID} = W_i^{(ID_i)}$ is an authentication function with computational complexity $O(1)$, which is deployed as a smart contract to the blockchain Genesis block.

### 3.1. Network Model

The mission environment of the UAV network contains multiple types of nodes: trusted nodes, which operate strictly according to the specified protocol and do not deviate from it in any way; untrusted nodes, which include external malicious nodes and internal error nodes that are dynamically generated during the mission, i.e., faulty nodes; UAVs that cannot complete incoming and outgoing messages; selfish nodes, UAVs that send data but do not forward network data; and compromised nodes, UAVs that delay, discard or even tamper with forwarded data. UAV networks in mission execution suffer from the Byzantine General problem and exhibit asynchronicity in most cases. Therefore, the trustworthiness of the UAV network depends on the current performance state of the UAV nodes participating in the mission, including the global reputation assessment of the nodes, the nodes' own residual energy and the number of neighbouring nodes. As the mission progresses, the credibility of the UAV nodes evolves. The blockchain trustworthiness system based on a

lightweight asynchronous provable consensus mechanism explored in this paper assesses the local state of the drone nodes in real time and achieves a consistent and up-to-date network-wide trustworthiness state assessment through the consensus of the authorisation committee to ensure high trustworthiness of the drone network in operation. Figure 1 shows the model of the unmanned network blockchain trustworthiness system during the mission execution phase.

All UAVs involved in the mission in the system model are full nodes of the blockchain system, and all UAV nodes are trusted at the beginning of the mission. Neighboring UAV nodes monitor each other's data forwarding behavior, while recording the number of their respective neighboring nodes. According to the set deduction rules, the drone nodes evaluate the local reputation of all neighboring nodes, collect the remaining energy, establish a record of the nodes' current trustworthiness status and multicast it to the delegated agents after the new block is chained. The Delegated Agent group is selected periodically and consists of the top M trusted nodes with the best status. The system authorizes the delegated agent group to perform consensus operations. The untrustworthy nodes are identified by counting the global state of the nodes, updating the delegated agents and finally creating new blocks.

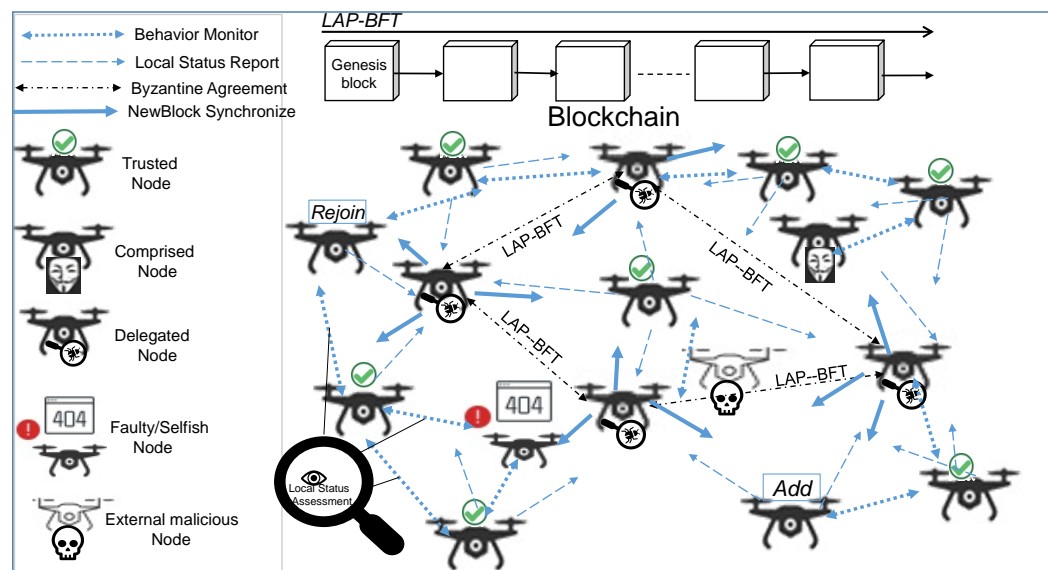

**Figure 1.** LAP-BFT consensus UAV blockchain network model.

### 3.2. Thread Model

In an unfriendly mission environment, external malicious nodes cannot only use their own powerful performance to carry out replay attacks, DOS attacks, etc., but also can take advantage of the openness of the wireless network to implement intrusions, such as link layer attacks, which cause legitimate UAV nodes to compromise and cause the UAV network to generate Byzantine nodes. As a special kind of mobile ad hoc network, the UAV network forwards data in a multi-hop, multi-path fashion via neighbouring node broadcasts. Therefore Byzantine error nodes, which are dynamically generated during the mission, cannot prevent the delivery of information between trusted nodes. However, errant nodes can discard messages sent by the correct node, or send inconsistent messages to different nodes, or deliberately delay the delivery of $U^i$ and $U^j$ messages from the correct node, tamper with the content of forwarded messages, misrepresent the state of the node itself, etc. Byzantine nodes can even collude with each other to improve the trustworthiness of each other's state. The selfish nodes in the error nodes' behavior of only receiving and not forwarding does not lead to malicious attacks, but it can also consume network resources. Too many error nodes in the system will not only seriously affect the overall

performance of the UAV network, but also cause the system to crash due to the inability of the consensus algorithm to complete.

### 3.3. Blockchain Strcture

UAVs act as blockchain nodes in a secure environment and register their identity, allocate public and private keys, generate a vector committed witness for authentication, etc. Initially, the system assigns a maximum global reputation value to all UAV nodes. The blockchain creation block is generated by a security center that is not involved in the mission. The Genesis block contains basic information about the registered UAV (a list of key–value pairs, $\{ID : (IP, PK, Reputaion)\}$. Identity vector commitments for all drones, the first round of delegated agents authorized to perform consensus operations, and the smart contracts required by the system are deployed in the Genesis block. The smart contracts mainly include a local reputation assessment smart contract, a node global status statistics smart contract and an identity authentication smart contract. The Genesis block is broadcast by the security centre to the network-wide nodes, and the drone nodes update the block chain. Each node activates the local instant status collection and evaluation function, monitors the data forwarding behaviour of neighbouring nodes, evaluates local reputation and collects the latest status. The delegated agent committee acts as the authorisation centre to collect and count the current trusted status assessment (global reputation of the node, remaining energy and number of neighbouring nodes) of the registered drone nodes. The delegated agent committee performs consensus on the collected local state data and the new delegated agent committee is finally elected, and the change committee will perform the next round of consensus operations. The block structure and blockchain form is shown in Figure 2.

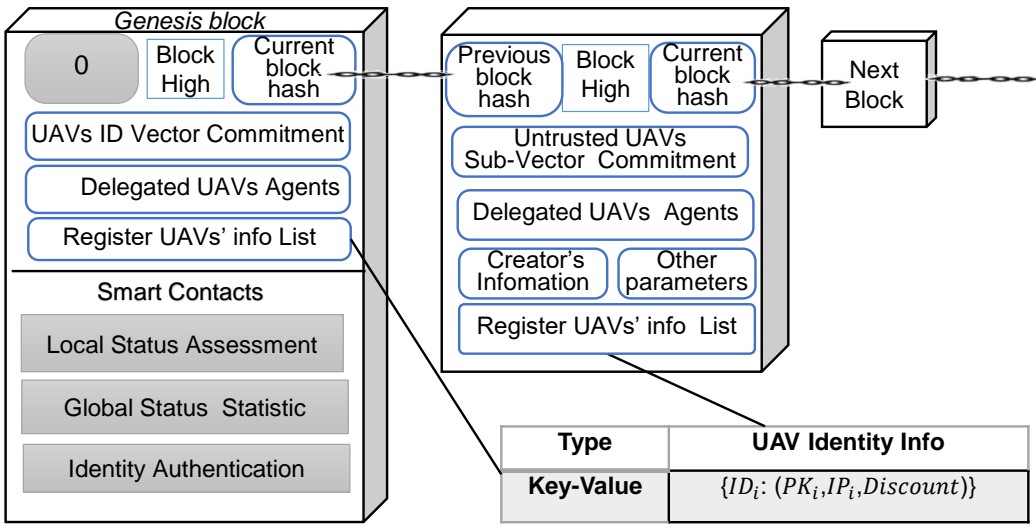

**Figure 2.** UAV network blockchain structure.

### 4. Recommended Solution

The lightweight asynchronous provable consensus Byzantine fault-tolerant consensus mechanism recommended in this paper operates as a blockchain in a drone network and is called a blockchain node state detector. All nodes monitor the data forwarding behavior of their neighboring nodes in real time and assess the behavioral trust discount of their neighbors' forwarding; delegated authorized nodes collect local state data for global reputation statistics and Byzantine fault-tolerant consensus, including decentralized reliable transmission, provable multivalue Byzantine consensus, global trustworthy state assessment, and blockchain synchronization after the creation of new blocks. The state detector continuously updates the global reputation assessment, providing the basis for the trusted operation of the UAV network. After a brief description of the UAV network trust system setup, this section describes the details of the UAV network trusted blockchain system

consensus algorithm in turn. Figure 3 shows the process of the dynamic maintenance of network trust by the enrolled nodes and periodically elected delegated agent nodes of the UAV network during the mission. The process includes the generation of node state transactions, trust consensus reaching, new block creation and blockchain updates.

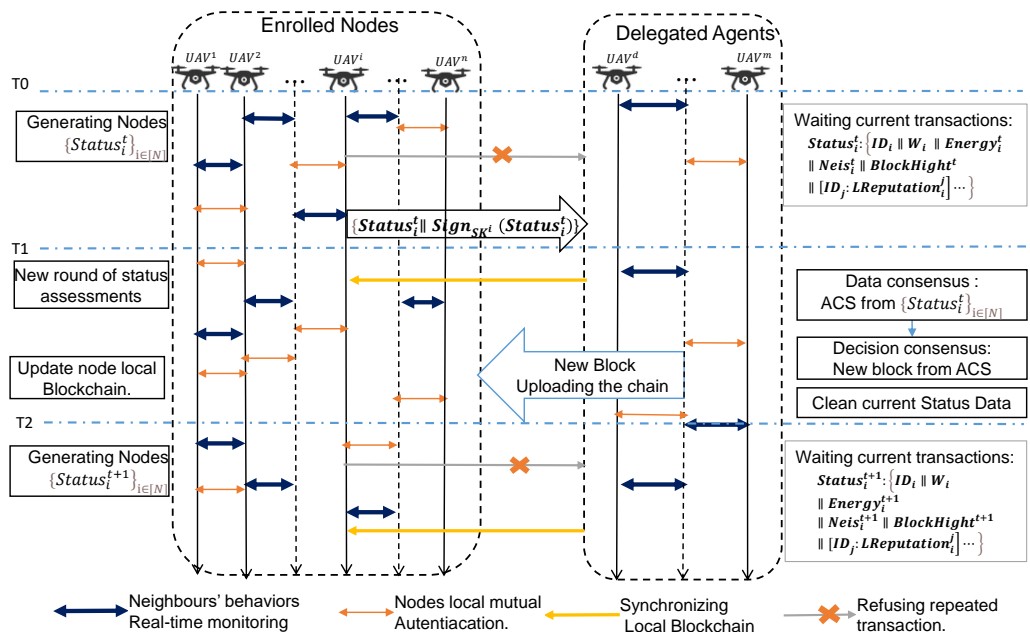

**Figure 3.** Flowchart of UAV network node status detection and consensus.

### 4.1. Node Trusted Status Detection: Local Trusted Status

All mission nodes implement local trust assessment by monitoring the forwarding behavior of their neighboring nodes at the network layer and collecting their latest operational status, mainly the node's remaining energy and the list of neighboring nodes. Nodes periodically multicast updated status packets containing the current blockchain height to the delegate agent. The assessment method is a reputation discount for bad behavior. Initially, all nodes are trusted with an initial reputation of 10.0. Nodes are identified as untrustworthy when their global reputation assessment is below equal to 0. The local state assessment algorithm classifies the forwarding behavior of UAV neighboring nodes into the following four types: normal forwarding, delayed forwarding, sending data but not forwarding data and forwarding incorrect data, and it penalizes them with a reputation discount. The discount scoring rules are shown in Table 2. For unresponsive behavior of faulty nodes, as well as collusive spoofing, local assessment is difficult to screen, but problems can be identified by statistical analysis of the consensus results.

**Table 2.** Classification of data forwarding behavior and penalty rules.

| Data Forwarding Behavior | Reputation Discount |
| --- | --- |
| Normal forwarding | 0 |
| Delayed forwarding | −1 |
| not forwarding data | −2 |
| Forwarding of incorrect data | −3 |

Two neighbor lists, (*Neis*) and (*Neis_check*), are set up to hold the two types of neighbor nodes detected in each round, one to hold the identity information ID of the neighbor node that requested to forward data, Algorithm 1 (line 4–7), and the other to hold the identity information of the neighbor node that requested to forward data after it

helped itself. The other type is a neighbor node that is monitored after helping itself to forward data, which saves the identity information ID of the neighbor node that requested to forward data and sets the local state collection period, the size of which is dynamically set according to each round of network state. The scheme is improved from the inter-zone protocol IARP in ZRP, where after detecting data returned by a neighbouring node, the node no longer discards the data directly, but reviews the content of the data. If the returned packet is forwarded out by the node, it looks for tampering, delayed message forwarding, and confirmation that the data has not been discarded by checking the list of neighbouring nodes. The Algorithm 2 is a local reputation evaluation function in which two timeout thresholds are set, i.e., the person counts its own broadcast data from time t1 and considers it to be intentionally delayed if it receives forwarded data beyond t1. The lists of two neighbouring nodes in the current round are compared to find out if there is selfish behaviour and to penalise it. Calculate the current trustworthiness loss of all neighbouring nodes according to the rules in Table 2. A normal forwarding discount score of 0 is given, 1 point is deducted if there is an intentional delay, 2 points are deducted if one sends one's own data and does not forward others' data, and 3 points are deducted if there is data tampering. Lines (13–20) of the algorithm refalg:lsc and lines (1–28) of the algorithm refalg:lsa implement this idea. The final tally is the discount score for each neighbouring node. Suppose $ID_x$ is a neighbouring node of $ID_i$, $CurScore_i^x$ denotes the reputation loss score for the current round, and $Discount_{(i-x)}^k$ is the discount estimate of $ID_i$ for the $k$th forwarding behaviour of $ID_x$. $k$ is the cumulative value indicating the number of times the forwarding behaviour of a neighbouring node has been detected in the current round. In this way the discount scores for all behaviours of the nodes at this stage are recorded and their average is used as the latest local plausibility loss.

Due to the nature of wireless networks, there is a conflict between receiving and sending, and it is possible for a node to fail to receive data returned by a neighbouring node and produce a misjudgement of the neighbouring node's behaviour in dropping forwarded data. Treating the local reputation assessment in the loop as an average can weaken the effect of misjudgement on the assessment. Also only a credibility discount penalty is applied to the behaviour, which effectively avoids high score assessments where malicious nodes collude with each other. Thus, at the end of a round, the discounted value of the local trust status of neighbouring nodes is as follows.

$$CurScore_{i-x}^{bcheight} = \frac{\sum_k^{nCount} Discount_{i-x}^k}{nCount} \tag{2}$$

where $nCount$ is the number of times $ID_i$ detected $ID_x$ forwarding behavior in the current round of detection. The local reputation evaluation of all neighboring nodes is denoted $CurScore_{i\ (x \in [Nei])}^x$, [Nei]neighboring nodes set. The node obtains the current energy value $Energy_i$, constructing a local reputation state record for the current round of $ID_i$.

$$LSA_i^t = \{ID_i \parallel W_i \parallel Energy_i \parallel Neighbors_i \parallel \\ curBlockHigh \parallel CurScore_{i\ x \in [Nei]}^x \parallel Sign_{SK_i}\} \tag{3}$$

and multicast to the delegated agents of the current round, where $W_i$ is existential witness, denoted as $ID_i$, existing in the identity vector commitment, and the authenticity of the data source is verified by the identity authentication smart contract in the Genesis block. $Sign_{SK_i}$ is the signature of $ID_i$ on the local state data used to ensure the integrity of the local trusted data. $curBlockHigh$ corresponds to the height of the blockchain, indicating the current round, and is used to prevent replay attacks. The process of collecting a round of local trusted data is completed by the function $buildLocStatusPack$ in Algorithm 2 and line (15–21) in Algorithm 1.

---

**Algorithm 1** LSC: Local Status Information Collecting.

---

**Let:** $Neis \leftarrow \{\}$, $Neis\_check \leftarrow \{\}$,
$Tc \leftarrow fixedV$.
**Let:** $curTurnScore \leftarrow \{\}$, $localDiscount[N][] \leftarrow 0$,
$count \leftarrow 0$, $expired \leftarrow 0$

1:　/* **Protocol for an UAV** $ID_i$ */
2:　**while** 1 **do**
3:　　**if** $Tc < 0$ **then**
4:　　　$Tc \leftarrow fixedV$
5:　　**end if**
6:　　$curtime \leftarrow getlocaltime()$
7:　　$curBlockHigh \leftarrow getcurBlockHigh()$
8:　　**while** true **do**
9:　　　**upon** received neighbor's packet, not from itself **do**
10:　　　**if** $ValidForwardingPacket(data\_j) = 1$ **then**
11:　　　　$Neis \leftarrow Neis \cup ID_j$
12:　　　**end if**
13:　　　**upon** $Boardcast(message^i, ID_i)$ **do**
14:　　　$count \leftarrow count + 1$
15:　　　$\{localdiscount[j], Neis\_check\} \leftarrow LocReAssess(locdiscount[j], Neis\_check)$
16:　　　**if** $curBlockHight = 0$ **until** $getlocaltime()\check{}curtime = Tc$ **do**
17:　　　 $expired \leftarrow 1$
18:　　　**if** $curBlockHight > 0$ **until** $getcurBlockHigh() > curBlockHigh$ **do**
19:　　　$expired \leftarrow 1$
20:　　　$curBlockHigh \leftarrow getcurBlockHigh()$
21:　　　**while** expired = 1 **do**
22:　　　　$localdiscount[j] \leftarrow localdiscount[j]/count$;
23:　　　　$curTurnScorecurTurnScore \cup \{ID_j \parallel laocaldiscount[j] \parallel Hash_j\}$
24:　　　　$curNeis \leftarrow getCurNeighbors(Neis, Neis\_check)$
25:　　　　$LSA_i^{curBlockHigh} \leftarrow buildLocStatusPack(curScore, curNeis, ID_i)$
26:　　　　$Neis \leftarrow \{\}, Neis\_check \leftarrow \{\}$
27:　　　　$curTScore \leftarrow 0, count \leftarrow 0, expired \leftarrow 0$
28:　　　　$multicast(Agents, LSA_i^{curBlockHigh})$
29:　　　**end while**
30:　　**end while**
31:　**end while**

---

*4.2. Node Trusted Detection, Global Trusted Status Assessment*

The current global trusted state assessment of all nodes is performed by a group of delegated agents authorized by the system. During mission execution, honest UAV nodes send the local state data collected during the round to all delegated agents in a multicast fashion. As a result, the delegated agent nodes receive the same node's local state data consistently, but there is no guarantee that the total local state data set is the same for all agent nodes. The reasons for this are mainly interference from errant nodes in the asynchronous UAV network, network partitioning, etc., resulting in inconsistent numbers of nodes completing data communication with different delegate agents at this stage. In order to obtain a consistent local state dataset, consensus must be reached between the delegate agents. At the same time, to ensure that the delegate agents are trustworthy, a set of trusted UAV nodes with the best state is selected to update the committee of delegate agents based on the consensus results. A consensus result is generated between all honest proxy agents, i.e., a public subset of the local trusted state dataset at this stage, and the global trusted state of all nodes is evaluated based on the consensus result, which is ultimately used to create new blocks and update the blockchain. The aim of this design is to avoid single point risks, balance the consumption of network resources, minimise the probability of errant nodes becoming proxy agents and, more importantly, to enable honest

proxy nodes to have deterministic and consistent outputs for all collected local trusted state data. The detailed procedure is described in algorithm refalg:lap-b. Let $ID_d$ be some proxy agent and the node state detector consensus operate in $ID_d$. At the beginning of the run, the list $S \leftarrow \{.\}$ is set to be empty to hold the identity ID of the sender of the local state data record; $Td$ is the maximum consensus period, taken from the current latest block and dynamically assigned according to the actual state of the consensus in each round. $Td$ is used to ensure that when the set of delegate agents has more than $Td$, it is used to guarantee the activity of the asynchronous consensus algorithm when more than one-third of the erroneous nodes are in the delegated agents set; $[M]$ and $[N]$ denote the current round delegated agents set and the set of all mission nodes, respectively. $RBCPacket_d$ denotes the subset of locally trusted state records from $ID_d$ used for consensus in the current round. The process of asynchronous consensus consists of three major steps: lightweight dispersed reliable transmission, provable multivalued Byzantine consensus and new block creation and blockchain updates. The Lightweight Dispersed Reliable Broadcast subprotocol.

The purpose of the Reliable Broadcast subprotocol (RBC) is to reliably transmit the proposed values proposed by each node to other nodes in the system. The proposed LD-RBC scheme builds on the traditional RBC protocol of Bracha [29], dropping the corrective coding (erasure code) scheme used in [8–10,29] with identity vector commitment authentication and data integrity verification instead of threshold encryption. Instead of randomly selecting encrypted transactions from a pool of transactions, the collected trusted state records are segmented according to the order of the delegated agent list and the length of the list to form dispersed packets of consistent length, reducing computation and improving transmission efficiency. Algorithm 3 starts the round by locally initializing the trusted state records of the registered nodes. The delegated agent $ID_d$ processes the collected local trusted state data, Line (7–12). First, verify the smart contract and hash function for legitimate validation, including identity authentication and data hash validation, based on the identity commitments of all registered UAVs in the node's local creation block, with a validation cost of $O(1)$. Secondly, check if there are duplicate data, as only the data collected in the current round is stored locally, it is easy to determine if it is the required data set for the agent based on the block height in the submitted data, and discard it if it is duplicate data to avoid replay attacks. If the block height in the received packet does not match the delegated agent block height and is higher than the agent block height, simply discard the process; if it is lower than the agent block height, perform blockchain synchronization on the sending node. After confirming that the received state data is valid and legitimate, it is saved to the local database. The node's ID that sent the trusted data is also saved to cache S, where $Td$ is the maximum duration of the collection record, which is stored as parameter information in the block structure and is dynamically calculated based on the actual communication status each round, a process represented by lines 16–19 in Algorithm 3. The dispersed packets are obtained by firstly calculating the size $B$ (number of state records) of the dispersed packet, e.g., line (20–22), $B \leftarrow (N/M)$, and secondly, by the position ordinal number of $ID_d$ in the list of delegated agents taking the number of locally trusted states in the corresponding position subset as the dispersed packet. Since there is no process of threshold encryption and decryption, the number of communication bits for $ID_d$ to submit data is $O(B|LSA_d|)$. Finally, the agent node $ID_d$ sends its own dispersed packet (Equation (4)) to the other agent nodes according to the RBC protocol.

$$RBCPacket_d = \{ID_d \parallel W_d \parallel LSA_d \parallel Sign_{SKd}(Td \parallel LSA_d)\} \tag{4}$$

---

**Algorithm 2** Functions with related Parameters for LSA.

---

**Function** *ValidateForwardingPacket*($data\_j, ID_i$)

1: Parse data_j as $\{ID_j \parallel W_j \parallel Message \parallel Sign\}$
2: /* $ID_i$ get $AC$ and $PK_j$ in its genesisblock */
3: $\{AC, PK_j\} \leftarrow getACandPKfromGenesis(ID_i, ID_j)$
4: **Output:** $VeriOpen(ID_j, W_j, AC)\&\&VeriSign(PK_i)$

**Function** *LocDiscountAssess*($curScore, Neighbors, ID_i$)

1: **let** $t0 \leftarrow v1(ms), t1 \leftarrow v2(ms), curScore_j \leftarrow 0$
2: $Brodcast(message_i)$#after $ID_i$ Broadcast Message
3: **let** $curtime \leftarrow getlocaltime()$
4: **let** $Discount_j \leftarrow 0; count \leftarrow count + 1$
5: **upon:** get back the message from the neighbors **do:**
6: /* parse the behavior of neighbors */
7: **if** $getlocaltime() - curtime < t0$ **then**
8:     **if** $message_{j-i} = message_i$ **then**
9:         $Discount_j \leftarrow 0$
10:     **end if**
11:     **if** $message_{j-i} \neq message_i$ **then**
12:         $Discount_j \leftarrow -3$
13:     **end if**
14: **end if**
15: **if** $getlocaltime() - curtime > t0$ **then**
16:     **if** $message_{j-i} = message_i$ **then**
17:         $Discount_j \leftarrow -1$
18:     **end if**
19:     **if** $message_{j-i} \neq message_i$ **then**
20:         $Discount_j \leftarrow -3$
21:     **end if**
22: **end if**
23: $Neighbors_{check} \leftarrow Neighbors_{check} \cup ID_j$
24: **if** $getlocaltime() - curtime > t1$ and $ID_j \in Neighbors$ **then**
25:     $Discount_j \leftarrow -2$ /*selfish behavior*/
26: **end if**
27: $curScore = curScore + Discount_j$
28: **Output** :$\{curScore, Neighbors\_check\}$

**Function** *buildLocStatusPack*($curScore, curNei, ID_i$)

1: $Energy_i \parallel getDevResidualEnergy(ID_i)$
2: $curBlockHigh \leftarrow getcurBlockHigh(ID_i)$
3: $status = ID_i \parallel W_i \parallel Energy_i \parallel curNeighbors \parallel curBlockHigh \parallel curScore$
4: $Mac \leftarrow Hash(status)$
5: **Output** :$\{status \parallel Sign_{sk_i}(status) \parallel Mac\}$

---

---

**Algorithm 3** LAP-BFT running in Delegated Agent $ID_d$.

---

/*[N]:UAV Nodes,[M]:agents */
**Let:** $S \leftarrow \{\}$, $Td \leftarrow 0$,$[M] \subseteq [N]$
/*refer to $|R|$ Records of RBC, $ID_d$ is the sender*/
**Let:** $LStatusPacket_d \leftarrow |R|$

1: **while** 1 **do**
2:     $InitlizeLocalDB(\perp)$
3:     $curTime \leftarrow getlocaltime()$
4:     **if** $isAgent(ID_d) = 1$ **then**
5:         $validwork \leftarrow 0$
6:     **end if**
7:     **while** true **do**
8:         $Td \leftarrow getAgentCollectfromCurBlock(ID_d)$
9:         **if** $stop = 1$ **then quit**
10:        /*$ID_x \in [N]$,$x \neq d$*/
11:        **upon** Receiving the $LSA(ID)$ **do**
12:        $S \leftarrow S \cup ID_x$
13:        **if** $ValidateLSA(ID_x) = 1$ **then**
14:           /*Locally save LSArecord*/
15:           $SaveLocalDB(LSA(ID_x))$
16:        **end if**
17:        **if** $curBlockHigh < getBlockHigh(ID_d)$ **then**
18:           /*Synchronize the local blockchain of $ID_x$*/
19:           $unicast(ID_x, sychronousBlocks)$
20:        **end if**
21:        **wait until** $Td$ is expired **do**
22:        **if** $|S| < (N+1)/3$ **then** $stop \leftarrow 1$
23:        **else**
24:        $Td \leftarrow (getlocaltime() \check{} curtime) \times (N/|S|)$
25:        $B \leftarrow (N/M)$
26:        $index \leftarrow getIndexOfAgentsList(ID_d)$
27:        $LSA_d \leftarrow getLocalDB(index, B)$
28:        $LStatusPacket_d = \{ID_d \parallel W_d \parallel Td, LSA_d \parallel Sign_{SKd}(Td, LSA_d)\}$
29:        Reliable Broadcast $[LStatusPacket]$
30:        $stop \leftarrow 1, validwork \leftarrow 1$
31:        **if** $stop = 0$ **then break** /* Collection is over.*/
32:     **end while**
33:     **if** $validwork = 1$ **then**
34:         $ValidAgents \leftarrow 0$
35:         $S[B][] \leftarrow \{\}$ /* B: RBCPackets number*/
36:         $Ts \leftarrow getMaxRBCtimefromCurBlock()$
37:         $curtime \leftarrow getlocaltime(), expired \leftarrow 0$
38:     **end if**
39:     **upon receiving** $|S[index]| = 2 \times M/3$ **do**
40:     $ValidAgents \leftarrow ValidAgents + 1$
41:     $ReplaceLocalDB(RBCPacket, index)$
42:     **upon** receiving $\{RBCPacket, index, Sign_{SK\_j}\}$ firstly **do**
43:     **if** $ValidRBCPacket(\{RBCPacket, index\}) = 1$ **then**
44:        $S[index] \leftarrow \{S[index] \cup \{ID_j \parallel Sign_S K_j\}$
45:        **deliver** $\{RBCPacket, index\}$ **to other agents**
46:     **end if**
47:     **upon** $ValidAgents = 2 \times M/3$ **do**
48:     get untrusted Nodes; updated Agent Nodes
49:     statistic agent's $LSA$; create own new Block
50:     uploading local blockchain
51: **end while**

---

### 4.3. Provable Multivalue Byzantine Consistent Subagreement (PMVBA)

After each agent node receives a dispersed packet from another agent, it verifies the identity of the agent node sending the packet and validates each record in the dispersed packet as legitimate through a smart contract in the Genesis block. First, the sender's identity is verified, and then the received data records are verified with records from the recipient's local corresponding region. If there are local state data records that are not identical, the sender is deemed to have tampered with the data and the data submitted by this agent node is rejected, otherwise the broadcast continues after adding its own signature. When the number of additional signatures is, for example, greater than two-thirds of the total number of delegated agents, $|M| * 2/3$, it indicates that a deterministic consensus result is obtained. At this point, the consensus time $Ts$ is counted, and $Ts$ is used for the next round of the maximum consensus cycle calculation. (The $Ts$ in the Genesis block is an empirical value obtained from experiments, in this paper, we use 100 nodes for experiments, set the existence of 10 faulty nodes and 20 Byzantine nodes, obtain the time required to complete a consensus $Ttest$ and calculate the maximum time for the first consensus by $Ts = N \times 3/10 \times Ttest \times 1.2$). Algorithm 3 starts at line 39 for the consensus phase, and line (47–49) demonstrate the process. The function $ValidRBCPacket$ proves the legitimacy of the packet, including authentication of the originating node to indicate that the data source is valid; the hash of the data content is verified to determine that the data has not been tampered with. Each delegate agent receives the decentralised packet, signed and added to the packet in the form of $ID_x \parallel Sign_{S_x}(Hash)$ after confirming validity with the above function. Verify all hash signatures to prove that the data has been signed and confirmed by multiple honest delegated agents. Algorithm 3 in line (43–46) adds the signature information to the corresponding cache, $S[B][]$, for all signature information of the dispersed packet, as in line 44, $S[index] \leftarrow S[index] \cup \{ID_x \parallel Sign(SK_x)\}$, and subsequently proceeds to send the dispersed packet line (40–42). When the number of valid signatures reaches two-thirds, the node status record in the local database corresponding to the serial number is replaced. When two-thirds of the dispersed packets are acknowledged, the consensus process ends and the extraction of the asynchronous common subset is completed. The rules for global reputation discounts are shown in Table 3.

**Table 3.** The rules for global reputation discounts.

| Reasons for Trust Discount | Global Discount |
| :---: | :---: |
| $Discount_{\sigma}^{x} > 5\sigma^{x}$ | $-0.5$ |
| Selfish Node, $Discount_{error}^{x}$ | $-1$ |

According to the rule, Equation (5) corrects the global trust discount of node $x$. Equation (6) calculates the current global trustworthiness.

$$GDiscount^x = GDiscount^x + Discount_{\sigma}^{x} + Discount_{error}^{x} \tag{5}$$

$$GReputation_x^{bcheight} = GReputaion_x^{bcheight-1} + GDiscount^x \tag{6}$$

When $GReputation_x^{bcheight}$ is less than or equal to 0, it means that the node is also untrustworthy and its node ID and its authentication information are accumulated to the wrong node subvector commitment and isolated from the network.

### 4.4. Delegated Agent Nodes Elections

The dynamic nature of the UAV network requires the upper-layer management network to be time-varying, and the nodes that make up the upper-layer network need to be not only trusted but also regionally representative. The upper-layer network constructed by the trusted central nodes of each subregion of the UAV network at each stage minimizes redundant routes. This solution uses clustering algorithms to regionally de-

lineate the UAV network and find the regional centers. The clustering feature is a list of neighboring addresses of the UAV nodes. The feature UAV node $i$ is represented as: $\overrightarrow{N_i} = [ID_{i1}, ID_{i2}, \cdots, ID_{im}]$, $m$ is the number of its neighboring nodes, and if the number of neighbors is less than $m$, the missing part is filled with zeros. The UAV network is represented by a feature vector as $\overrightarrow{U} = \left[\overrightarrow{N_1}, \overrightarrow{N_2}, \cdots, \overrightarrow{N_n}\right]$, with $n$ being the number of UAVs in the current mission network. The feature vectors used for clustering are not of numerical type but are lists used for classification. Therefore, a fuzzy K-modes clustering algorithm [30–32] is used, replacing the mean with the mode as the central node of the zone (cluster) and adapting to the situation of overlapping regions. Clustering makes use of simple matching dissimilarity, i.e., the dissimilarity between two UAV nodes is expressed in terms of the cumulative number of m neighbors of the feature vector, the fewer the mismatches, the closer the two nodes are. The mathematical expression (7) shows the proximity of two UAV nodes.

$$d(\overrightarrow{N_i}, \overrightarrow{N_j}) = \sum_{x=1}^{m} \sum_{y=1}^{m} \delta(ID_{ix}, ID_{iy}) \tag{7}$$

where

$\delta(ID_{ix}, ID_{iy}) = 0$, if $(ID_{ix} \neq ID_{iy})$;
$\delta(ID_{ix}, ID_{iy}) = 1$, if $(ID_{ix} = ID_{iy})$.

Let $U^k = [\overrightarrow{N_1^k}, \overrightarrow{N_2^k}, \cdots, \overrightarrow{N_n^k}]$ be a subzone of the UAV network, and the mode in the UAV network denotes the feature vector of the central node of the zone.

**Definition 1.** *The feature vector $Q = [ID_1, ID_2, \cdots, ID_m]$ is the mode of the UAV network $U^k$, if it makes the function (8),*

$$D(Q, \overrightarrow{N_i}) = \sum_{i=1}^{m} d(\overrightarrow{N_i}, Q) \tag{8}$$

*take the minimum value and $Q \in U^k$.*

Let $n_{ID_x}$ be the times the neighbor node $ID_x$ appears in all lists of neighbors, the frequency of $ID_x$ in the zone $U^k$:

$$f(ID = ID_x | U^k) = \frac{n_{ID_x}}{m} \tag{9}$$

**Theorem 1.** *Mode update method for k-modes of UAV networks; the function $D(Q, N_i)$ reaches a minimum when and only when the following inequality holds:*

$$f\left(ID = ID_x \middle| U^k\right) \geq f(ID = ID_j | U^k) \tag{10}$$

*where $ID_x \neq ID_j$, $\forall j = (1, 2, \cdots, m)$. After the upper layer network nodes reach consensus on the local state data at this stage, a consistent subset of common local state data is generated at this stage and the global trustworthiness of the nodes is tallied. The clustering process obtains the central nodes of the partition and uses these nodes as delegated agent nodes to form a new upper layer network for the next stage of consensus operation and inter-zone route discovery.*

The entire consensus process includes the detection of node trusted states, asynchronous consensus of local trusted state data by proxy, output of asynchronous common subset, then decision consensus on data consensus results, final determination of global trustworthiness of nodes, and new members of proxy members. Finally, the updated blockchain provides a trusted foundation for the continued operation of the UAV network, one round of consensus for LAP-BFT can be summarized in Figure 4.

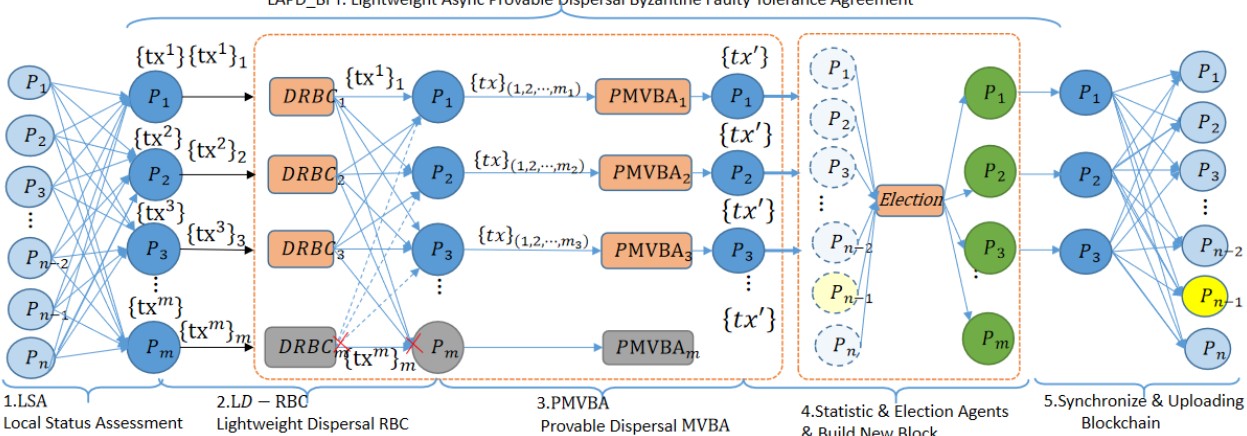

**Figure 4.** One-round consensus process for the LAP-BFT protocol.

A mission network of n UAVs (light blue nodes). All mission nodes send the latest local trusted state data to the delegate agents (m blue nodes), the delegate agents receive a local state dataset $tx_{i \in \{1,2,\ldots,m\}}^i$ that is not guaranteed to be consistent, generate a consistent local state dataset $tx'$ via LAP-BFT, honest delegated agents statistics $tx'$, elect a new set of delegated agents (a set of green nodes), identify untrustworthy nodes (yellow nodes), create a new block based on the statistics and broadcast it across the network to update the blockchain.

## 5. Proof of System Properties and Performance Analysis

The ultimate design aim of this solution is to achieve atomic broadcast of the latest state assessment of UAV nodes in an asynchronous UAV network and to establish consistent networkwide block data of the global section trusted state to support continuous trusted running of the UAV network. Formally, an atomic broadcast protocol satisfies the following properties with overwhelming probability.

- Agreement, if an honest node outputs a value $v$ then every honest node outputs $v$.
- Total Order, if two honest nodes output $\langle v_0, v_1, \ldots, v_j \rangle$ and $\langle v'_0, v'_1, \ldots, v'_j \rangle$, respectively, then $v_0 = v'_0, v_1 = v'_1, \ldots, v_j = v'_j$.
- Censorship resilience, if a value $v$ is input to $n - f$ honest nodes, then it will eventually be output by each honest node.

The Lightweight Asynchronous Provable Byzantine Fault-Tolerant Consensus Protocol (LAP-BFT) is an asynchronous common subset (ACS) consisting of the Decentralized Reliable Broadcast Subprotocol (DRBC) and the Provable Multi-Valued Byzantine Consistency Subprotocol (PMVBA). Combined with the smart contract used for validation in the Genesis block, it can be efficiently and simply converted to an atomic broadcast for all node local state datasets with the correct delegate agent node outputting the public subset. LAP-BFT MUST satisfy the Agreement, Total Order and External-Validity properties for security, and the Termination property for activity.

### 5.1. Proof of Security and Activity

**Theorem 2.** *The system satisfies the following Agreement property: At the end of a round of consensus protocol, if there exists an honest node that outputs a locally trusted set of state records $\{LStatusPacket_d\}_{(|R|)}$, then all honest nodes output $\{LStatusPacket_d\}_{(|R|)}$.*

**Proof.** $M$ delegated agents arrange the collected local trusted status data records ($LSAs$) into the dataset $\{LSA\}_N$ in the order of the nodes' identity vector commitments in the Genesis block. Disperse them into $M$ subsets of records of size $B = N/M$. The delegated agent nodes take the corresponding $B$ records in the order of their position in the

current delegated agent list, e.g., if the position of the agent node $ID_d$ requires Index, then $LStatusPacket_d = \{LSA\}_{(index+B)}$. After RBC communication, there are $R$ sets of $LStatusPacket_d$ data records in R honest nodes, which are combined into a consistent data set $\{LStatusPackeT_d\}_{(|R|)}$ in the order of Index size and proven. $\square$

**Theorem 3.** *The system satisfies the following Total order property: If an honest node outputs a sequence of messages $\{v_0, v_1, \ldots, v_j\}$, another honest node outputs a sequence of messages $\{v'_0, v'_1, \ldots, v'_j\}$, then $v_0 = v'_0, v_1 = v'_1, \ldots, v_j = v'_j$.*

**Proof.** The local trusted state dataset collected by the delegated agent group is ordered based on the order in which the identity vector commitments were generated at the time of UAV registration. Honest nodes transmit dispersed packets via a reliable broadcast protocol, which are then verified by a provable multivalue agreement protocol and finally concatenated according to the order of the delegated agent nodes' positions in their lists, so that the order of the data in all honest nodes is consistent and proven. $\square$

**Theorem 4.** *Corresponding to the resilience of censorship in atomic broadcasting, ACS requires verifiability of consensus results, and this scheme provides external validation. The system satisfies the following External-Validity property, such that if an honest node outputs a value $v$, then $Ext\_Verify(v) = true$, where $Ext\_Verify$ is external verification.*

**Proof.** The UAV network runs in an authenticated environment with an authentication function stored in the Genesis block as a blockchain smart contract. This includes authentication of the data sender; node data integrity verification, i.e., the node performs a $HASH(curScore_i^x)$ process on the reputation assessment values of its local data against neighboring nodes, and during the consensus process, the agent nodes receiving the dispersed packets compare the corresponding hash values to determine whether the sender has tampered with them. The legitimacy of the data sent by the honest node is confirmed by the data legitimacy verification smart contract provided by the blockchain, which is proven. $\square$

**Theorem 5.** *The system satisfies the following Termination property. Let $f$ be the number of error nodes in the delegated agent nodes. If $(f + 1)$ activates the PMVBA protocol and all messages between honest nodes (trusted delegated agents) arrive, then the honest nodes output $\{LStatusPacket_d\}_{(|R|)}$, where $|R|$ is the number of honest nodes.*

**Proof.** The PMVBA protocol is executed in all delegated agent nodes, and if there are $f + 1$ primary agents that are trusted honest nodes, then there are honest nodes that receive $|R|$ signed and acknowledged copies of the scattered packets, where $|R| = f + 1$, and eventually the honest nodes output $\{LStatusPacket_d\}_{(|R|)}$. However, even if $f > (N + 1)/3$, ($N$ is the total number of nodes in the UAV network), the consensus protocol can terminate execution as long as the number of erroneous nodes in the agents $f' < (M + 1)/3$, ($M$ is the total number of agents). The UAV network operates in a complex environment, and it is possible that the agent nodes may not be able to meet the Byzantine fault tolerance requirements, resulting in a situation where the consensus protocol cannot be terminated. Because the agent has periodically elected trusted nodes to act as such, the data record can be verified between honest nodes, and therefore a consensus cycle can be set to resolve the activity problem. If the timeout fails to terminate consensus, but the number of scattered packet signatures has exceeded two, the master node is determined based on the order of the size of the agent ID and the local trusted state dataset of that master node is used as an asynchronous common subset to maintain consensus activity. Obtained. $\square$

### 5.2. Other Security Analyses

Preventing replay attacks: The UAV network blockchain system provides authentication based on identity vector commitment, where malicious external nodes in the mission environment cannot perform unauthorized access but can still launch replay attacks. The height of the current block is included in the local trusted state packet, while the agent node saves one round of collected local trusted state data to local data, so that duplicate data for the same round is discarded directly. Data from different rounds are also rejected because they do not match the height of the local block chain. This effectively prevents replay attacks, and the same round of data can be cleared after a new block is chained, avoiding storage pressure.

Preventing erroneous blocks from being chained: Honest delegated agent nodes in the consensus process locally validate the received dispersed trusted status packets, validated as described in Theorem 4. If the delegated agent transforms into a Byzantine node, the validation of the dispersed packets it sends does not pass. Add the signature $Sign_{SKx}(LStatusPacket_d)$ to verify legitimacy. If passed, the signature is merged into $S$ as a key-value pair $ID_x : (Sign_{SKx}, LStatusPacket_d)$. The UAV node receiving the new block prevents forgery of it by a malicious node in the delegate agent by verifying $S$.

### 5.3. Effectiveness Analysis

The completion of missions in complex environments with resource-constrained UAV networks relies on how their systems meet lightweight requirements. This scheme provides a lightweight asynchronous provable multivalued Byzantine consensus algorithm for the analysis of UAV networks in terms of communication, computation (energy consumption) and storage.

Lightweight communication: The scheme provides de asynchronous consensus algorithm where consensus transactions are local state records generated by the nodes themselves at this stage, rather than transaction data from third parties. Primarily at the network layer neighbouring nodes collect data by monitoring each other's forwarding behaviour and performing reputation discount assessment. The message complexity depends on the number of one-hop neighbouring nodes of the node. In terms of the reputation discount evaluation value of a neighbouring node $ID_x, curScore_i^x)$, its size is 5 bytes, and even if the node $ID_i$ has 100 neighbouring nodes, its local state data length is much less than 1 KB. Let the single local status be provided by the node. The size of the data is $L$. Additionally, in this phase, multicast is sent to the proxy and the communication complexity is $O(1)$; in the consensus phase, the number of agent nodes, $M$, is much smaller than the number of nodes participating in the mission, $N$. This dispersed data is used for stable transmission (DRBC), and the length of the dispersed packet $|m| = L \times M/N$. The communication complexity of the synchronization phase is $O(M^2|m|)$, $Mgg\{N$ and there is no need to keep historical transactions in the block. The main component of the consensus data is the global reputation evaluation $(ID_i, Greputaion)$ of the current round of $N$ drone nodes, which is also 5 bytes. Thus even at a network size of 1000 drones, the new block size is only $K$ levels. The communication complexity of delegating the agent to broadcast the new block to the current network is $O(1)$.

Lightweight computation: The consensus algorithm provided by this scheme adopts the method of external proof function to verify the consensus result, the verification method does not need to traverse the query, such as identity authentication using vector commitment to verify the existence witness provided by the node, integrity verification by comparing the hash value and verifying the signature, etc.; the computational complexity is all $O(1)$. The output Asynchronous Common Subset (ACS) is a direct merging of the dispersed trusted state records submitted by the honest delegated agent nodes, which does not require the use of corrective codes to disperse and recover data because of the guarantee of external proofs, avoiding the computation of [8–10] threshold encryption and decryption. In addition, the scheme uses delegated authority to select the node with the best state for

each round of consensus computing and dynamic change, which also effectively achieves computational balance and extends the overall running time of the UAV network.

Lightweight storage: The UAV network trusted system is essentially a stateless blockchain, replacing the identity registration record with a 32-bit vector commitment. There is also no need to keep dynamically generated local trusted state data as a historical record. At each consensus stage, only the aggregated subvector of untrustworthy node witnesses from the global trusted state record and the list of new proxy agents are kept. The delegated agents only provide the local state data storage required for a consensus round. No cumulative storage is required.

### 6. Simulation Experiments

The experiments were conducted using the QualNet network simulation environment. The QualNet simulator, developed by Scalable Networks Technologies (SNT) for network design, operation and management, simulates the network behavior and performance of thousands of nodes and is a comprehensive set of tools for simulating large wireless or wired networks. The biggest difference between the UAV network trusted system and other application scenarios is that transactions are node-related, and consensus transactions are the state values of all nodes during runtime, rather than third-party customer-submitted data. The more nodes there are the more transactions there are. The UAV network has no error nodes at the beginning of the mission, and as the mission progresses creates faulty nodes, selfish nodes and Byzantine nodes.

Scenarios with different scales of wireless communication are set up in the QualNet simulation environment with a certain percentage of error nodes. The dynamic changes in the operation of the UAV network mission are simulated by configuring multiple sets of configuration files and by the node application importing different configurations at different time periods. Several asynchronous consensus protocols are deployed into the node stack separately, and the consensus process throughput, latency and energy consumption are compared for different combinations of mission nodes of different sizes and containing different proportions of error nodes.

In this scheme, the final asynchronous generic subset of ACS is generated by executing DRBC and PMVBA subprotocols. Let the length of the data to be consented $|m|$ and the simulation experiments focus on the message complexity, communication complexity and computational complexity in the related algorithms during the consensus process. Comparing the throughput, latency and computational overhead of the asynchronous consensus algorithms in [8–10] in one round of consensus. Table 4 shows a comparison of the performance of processing ACS.

**Table 4.** Complexity comparison for ACS.

| Protocol | Computation | Communication | Message | Fault Tolerance |
|---|---|---|---|---|
| HB-BFT | $O(\log^N)$ | $O(N^2|m| + \lambda N^3 \log^N)$ | $O(N^3)$ | 1/3 |
| Dumbo1 | $O(\log^k)$ | $O(N^2|m| + \lambda N^3 \log^N)$ | $O(N^3)$ | 1/3 |
| Dumbo2 | $O(1)$ | $O(N^2|m| + \lambda N^3 \log^N)$ | $O(N^3)$ | 1/3 |
| LPA-BFT | $O(1)$ | $O(N^2|M|)$ | $O(M^3)$ | $\geq$1/3 |

Where $\lambda$ is the length of the security parameter, $k$ is the number of fixed agents elected by Dumbo1 and $M$ is the length of the Delegated agents list.

The base latency: A round of consensus latency, is defined in this scenario using the latency in HB-BFT; the average time interval from the first node starting the protocol to the n-fth node obtaining the result. The latency is related to the size of the transaction volume and the number of participating nodes, and the application scenario in this paper, where the transactions are the nearest trusted state data of the nodes, and thus the more nodes the larger the transaction volume. The experimental design sets different network sizes and

configures no more than one-third of the total number of erroneous nodes in the consensus delay during a consensus round, and the experimental results are shown in Figure 5.

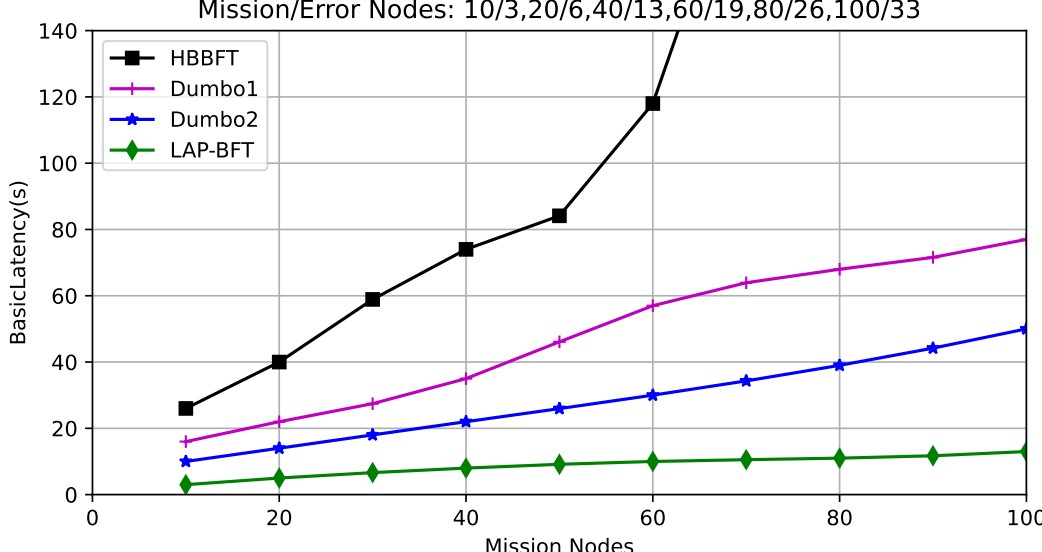

**Figure 5.** Comparison of consensus base delay.

It is obvious that the asynchronous consensus latency of HB-BFT is much higher than other schemes, mainly because the ABA subprotocol in the ACS protocol has multiple instances in each node, which not only consumes a large number of operations but also increases the consensus latency. In contrast, the algorithm in [9] uses an agent approach to reduce the number of ABA instances, but because the consensus result of the final generated ACS is achieved by all participating nodes randomly selecting a set of transactions using the RS_Code technique, the threshold encryption and decryption operations are added to the consensus algorithm, while in this case the asynchronous consensus algorithm of LAP-BFT only runs on the selected proxy and provides smart contracts through the blockchain for external proof to achieve the final ACS consensus result, which takes less time to run.

Latency under different error nodes: Multi-round consensus delay experiments to design application scenarios for a UAV network with 60 nodes. A different number of error nodes are generated in each round as the mission progresses. The number of error nodes is gradually increased in each consensus round by the UAV network through several sets of aggregate profiles, which are loaded into the system at different stages to simulate the dynamic generation of error nodes in a complex task environment. Comparing the latency of the four consensus algorithms is shown in Figure 6. It can be seen that the consensus algorithms of HB-FBT and Dumbo-HBT have a gradual increase in latency as the number of errant nodes increases, and the latency rises more rapidly. When the number of erroneous nodes exceeds 1/3 of the total number of participating nodes, the consensus reaching condition cannot be satisfied, the consensus process cannot be terminated and the latency is infinite. lAP-BFT uses dynamic selection of the best group of nodes based on reputation as the proxy for each round of asynchronous consensus to minimize the possibility of erroneous nodes appearing in the delegated agent nodes. This way the probability of more than 1/3 error nodes arising in the delegated agent nodes is very small. As a result, LAP-BFT asynchronous consensus satisfies the terminable condition even with erroneous nodes that exceed Byzantine tolerance, and with little change in consensus latency as erroneous nodes increase.

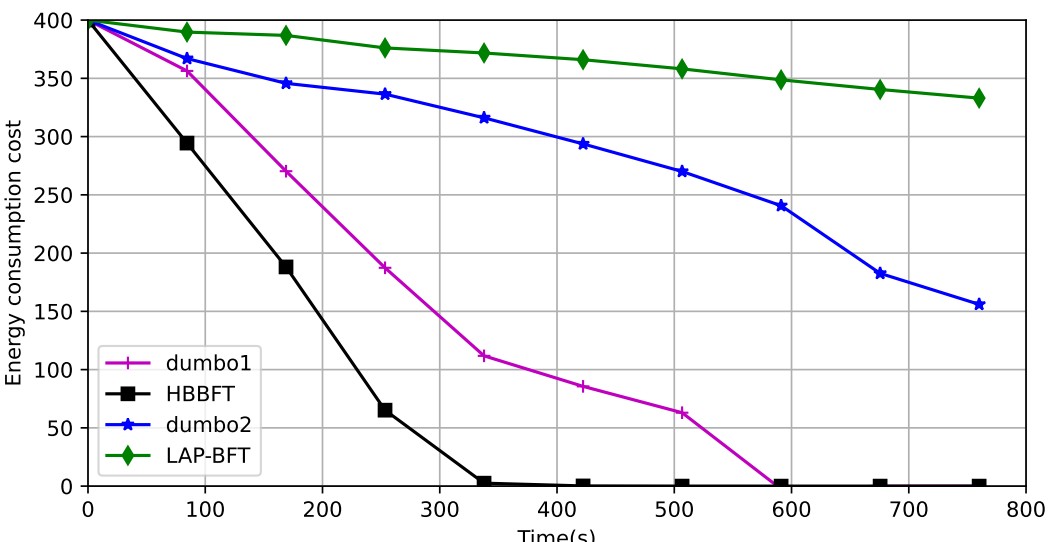

**Figure 6.** Comparison of delay times with different error nodes.

Relationship between throughput and latency: Throughput is the number of transactions submitted by the system per second and is a concept that is closely related to bandwidth. The transaction volume in the application scenario of this paper is related to the number of nodes, but the transaction data consists mainly of reputation discounts from neighboring nodes, which are small in order of magnitude. As the network size gradually increases, the node throughput also gradually increases, and after reaching a peak, the throughput decreases. Experiment-1 demonstrates that ABA is too consuming in the HB-BFT scenario and that the computational power of HB-BFT asynchronous consensus bottlenecks at more than 60 nodes. In addition, it is more meaningful to study the relationship between throughput and latency for this application scenario, so the experiments were designed for different scales of up to 60 drones, first to ensure that each asynchronous algorithm can run in these scenarios, and then for the consensus latency case at different throughputs; the maximum tolerable error nodes exist for each scale. The experimental results are shown in Figure 7. Again due to the different ways of building ACS, Dumbo1 and Dumbo2 require threshold signature encryption, and latency rises faster with throughput, while HB-BFT needs to handle threshold key processing, and the number of ABA instances per node increases with the number of nodes, and the number of nodes is proportional to the volume of transactions, so HB-BFT rises fastest with the number of nodes. Latency growth for LAP-BFT is more moderate because the computational complexity of external validation of the consensus process is constant and because there is only one instance of PMVBA per node and the computation occurs in a small number of delegated agents.

Rate of energy consumption: The UAV network's energy supply is limited, and extending the runtime of the UAV network is also an important manifestation of lightweighting. The verification environment of a UAV network of 50 registered UAVs generates 14 erroneous nodes at some stage. The asynchronous state of the network is simulated by setting node property parameters and specifying their forwarding behavior in the program run. Five faulty nodes, which do not process forwarded data; five Byzantine nodes, which randomly tamper with forwarded data; and four selfish nodes, which send data but do not forward it. The drones move on a given path, without considering obstacle avoidance. The simulated energy values are set to correspond to the running times of the different required algorithms corresponding to the nodes deployed on them, and the experiments compare the rate of consumption of a given amount of energy by different asynchronous consensus algorithms. This is shown in Figure 8:

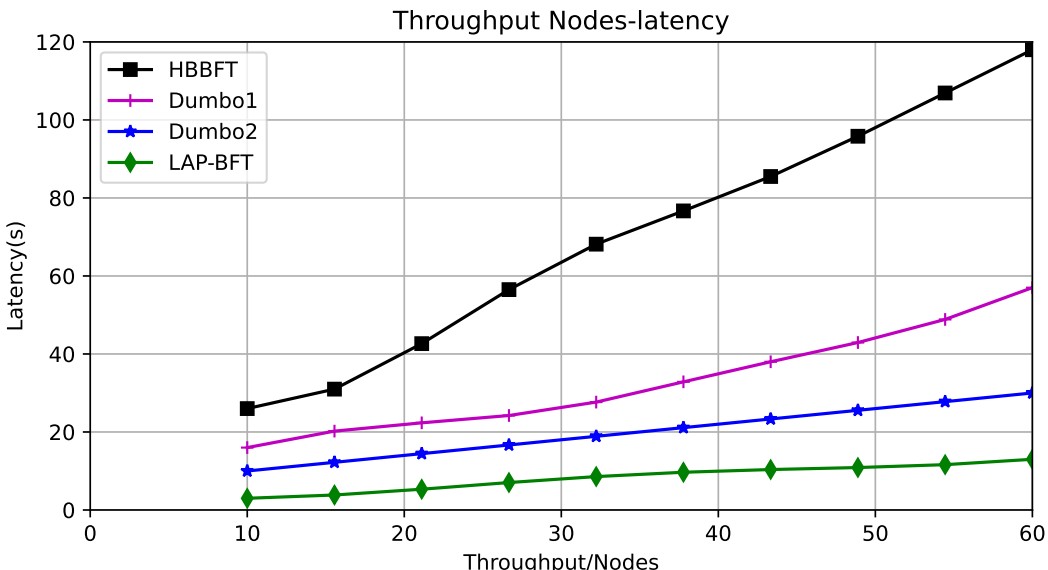

**Figure 7.** Throughput versus latency.

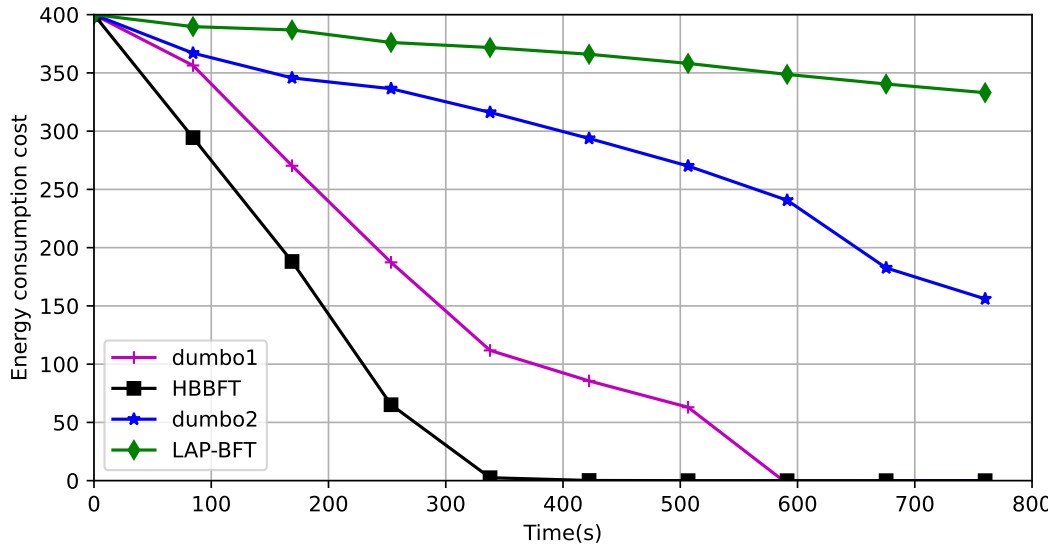

**Figure 8.** Comparison of energy consumption rates.

Dumbo1 uses fixed proxies and its computational complexity is related to the number of proxies, but because the consensus algorithm runs all the time, the proxy nodes are consumed quickly and the consensus process cannot continue when the proxy consumption ends. Dumbo2 uses more energy than LAP-BFT because the threshold key is still computed. This is because LAP-BFT uses only external verification to prove the consensus result, and more importantly, dynamically selects groups of delegated agents to share the consensus computation, extending the running time of the entire network.

## 7. Conclusions

The objective of this solution is to establish an asynchronous fault-tolerant system to maintain the trustworthiness of the UAV network during mission execution. Through mutual monitoring between drone nodes during data delivery, the nodes evaluate the behavior of their respective neighboring nodes and collect the latest current local trusted state. A lightweight asynchronous provable consensus is used to reach networkwide agreement on the global trusted state of the nodes, providing a trusted environment for the next round of drone network operation. The security and activity of the asynchronous consensus proposed in the scheme are explained in terms of theoretical proofs, and the

transmission efficiency and computational overhead of three practical asynchronous consensus algorithms operating in the UAV network environment are compared by QualNet network simulation software. Experiments with one or more rounds of consensus process show that this scheme has superior performance in terms of throughput, consensus latency and consensus computational overhead of a single round. The smart contract is used in many aspects of the scheme, such as authentication, proof of consensus results, etc. In an asynchronous environment, how to execute the smart contract dynamically according to the actual needs is the focus of the next paper.

**Author Contributions:** Conceptualization, L.K.; Data curation, L.K.; Project administration, B.C.; Writing—original draft, L.K.; Writing—review & editing, F.H. All authors have read and agreed to the published version of the manuscript.

**Funding:** This work was supported in part by the National Key Research and Development Program of China, under Grant 2019YFB2102002; in part by the National Natural Science Foundation of China, under Grant 62176122, 62001217; in part by A3 Foresight Program of NSFC, under Grant No. 62061146002.

**Institutional Review Board Statement:** Not applicable.

**Informed Consent Statement:** Not applicable.

**Data Availability Statement:** Not applicable.

**Conflicts of Interest:** The authors declare no conflict of interest.

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
