# Peer review of "LAP-BFT: Lightweight Asynchronous Provable Byzantine Fault-Tolerant Consensus Mechanism for UAV Network"

_drones, doi:10.3390/drones6080187_

Round 1
Reviewer 1 Report
This paper deals with a timely topic of fault-tolerant consensus mechanisms for UAV networks. As UAV networks become widespread in a variety of different domains, new age consensus mechanisms are needed that are lightweight and adaptable to changing network conditions.
1. Very well-written paper with comprehensive background information, problem definition, and solution approach backed with simulation and analytical results.
2. Minor English issues like castro -> Castro in line 56, A blockchain -> a blockhain in line 98.
Author Response
Comment 1 Very well-written paper with comprehensive background information, problem definition, and solution approach backed with simulation and analytical results
Response. . Thank you very much for your recognition of our work.
Comment 2 Minor English issues like castro -> Castro in line 56, A blockchain -> a blockhain in line 98
Response. According to the comment, We have carefully checked the whole document several times and fixed a number of errors similar to those mentioned in the Comment.
Reviewer 2 Report
To address the resource-limited nature of UAV networks, this paper proposes a lightweight asynchronous provable Byzantine fault-tolerant consensus method. The consensus method reduces the communication overhead by splitting the set of local trusted state transactions and then dispersing the reliable broadcast control transmission (DRBC), introduces vector commitments to achieve multi-value Byzantine consensus (PMVBA) for identity and data in a provable manner, reduces the computational complexity, and the data stored on the chain is only the consensus result (global trustworthiness information of the drone nodes), avoiding the Blockchain’s "storage inflation" problem. This makes the consensus process lighter in terms of bandwidth, computation, and storage, ensuring the longevity and overall performance of the UAV network during the mission.
The introduction and related work sections are well written and the authors explained the details. However, I think they can also include some references for phase synchronization of UAVs. For instance, these are two examples:
M. Mozaffari, W. Saad, M. Bennis, Y.-H. Nam, and M. Debbah, ‘‘A tutorial on UAVs for wireless networks: Applications, challenges, and open problems,’’ IEEE Commun. Surveys Tuts., vol. 21, no. 3, pp. 2334–2360, 3rd Quart., 2019.
A. Pourranjbar, M. Baniasadi, A. Abbasfar, and G. Kaddoum, ‘‘A novel distributed algorithm for phase synchronization in unmanned aerial vehicles,’’ IEEE Commun. Lett., vol. 24, no. 10, pp. 2260–2264, Oct. 2020.
In part 4, I suggest that the authors divide figure 3 into several parts and clearly explain each part since, in this current version, it is not easy to understand what is going on there exactly. Even the texts in this figure are not readable easily.
Algorithms 1,2, and 3 look promising. I am wondering if the authors have some discussions about the convergence of these algorithms theoretically. Are there any sufficient and necessary conditions under which these converge?
I like the complexity analysis in table 3. I think a similar table can be prepared for the convergence speed of the algorithms as well.
In general, this paper looks good and I can suggest it be accepted if the authors address my minor above-mentioned comments.
Author Response
Please pay attention to the Word Documnet.
